# Mothers' hygiene experiences in confinement centres: A cohort study

**Siew Cheng Foong**[ID]◉*, **Wai Cheng Foong**[ID]◉, **May Loong Tan**[ID]◉, **Jacqueline Judith Ho**◉

Department of Paediatrics, RCSI & UCD Malaysia Campus (formerly Penang Medical College), George Town, Penang, Malaysia

◉ These authors contributed equally to this work.
* scfoong@rcsiucd.edu.my

**Data Availability Statement:** The minimal anonymized data set has been uploaded as Supporting information.

**Funding:** RCSI & UCD Malaysia Campus provided a research grant (PMC RC-17) but did not play any

## Abstract

### Introduction

Ethnic Malaysian Chinese used to observe the 1-month postpartum confinement period at home and many families would engage a traditional postpartum carer to help care for the mother and newborn. A recent trend has been the development of confinement centres (CCs) which are private non-healthcare establishments run by staff not trained in health care. Concerns about hygiene in CCs arose after infections were reported. We describe the practice of hand hygiene observed in CCs, the availability of resources for hygiene, and the prevalence of health-related problems in CCs.

### Methods

This is a cohort study of ethnic Chinese mothers intending to breastfeed their healthy infants. They were recruited post-delivery along with a comparison group who planned to spend their confinement period at home. After their 1-month confinement period, they were contacted for a structured telephone interview about their experience. To avoid any alteration in behaviour, mothers were not told at recruitment that they had to observe hygiene practices. Multiple logistic regression was used to assess the effect of place of confinement on rates of infant health problems.

### Results

Of 187 mothers, 88(47%) went to 27 different CCs while 99(53%) stayed at home. Response rates for the 1-month interviews were 88%(CC) versus 97%(home). Mothers in CC group stayed in one to four-bedded rooms and 92% of them had their baby sleeping separately in a common nursery described to have up to 17 babies at a time; 74% of them spent less than six hours a day with their babies; 43% noticed that CC staff had inadequate hand hygiene practices; 66% reported no hand basins in their rooms; 30% reported no soap at hand basins; 28% reported inexperienced or inadequate staff and 4% reported baby item sharing. Among the mothers staying at home, 35% employed a traditional postpartum carer for her baby; 32% did not room-in with their babies, but only 11% spent less than 6 hours a day with their babies. Of mothers who employed traditional postpartum carers, 32% did not

role in the design of the study, data collection, analysis and interpretation of data nor in the writing of the manuscript or decision to publish.

**Competing interests:** The authors have declared that no competing interests exist.

**Abbreviations:** CC, Confinement centre; HPRI, Health problems that are probably related to infections; HPUI, Health problems that are unlikely related to infections.

know if their carer washed hands after changing diapers and 18% reported that their carer did not. Health problems that were probably related to infection (HPRI) like fever and cough were similar between the groups: 14%(CC) versus 14%(home) (p = 0.86). Multiple logistic regression did not show that CCs were a factor for HPRI: aOR 1.28 (95% CI 0.36 to 4.49). Three mothers reported events that could indicate transmission of infection in CCs.

## Conclusion

We found unsatisfactory hygiene practices in CCs as reported by mothers who spent their confinement period there. Although we were not able to establish any direct evidence of infection transmission but based on reports given by the mothers in this study, it is likely to be happening. Therefore, future studies, including intervention studies, are urgently needed to establish an appropriate hygiene standard in CCs as well as the best method to implement this standard. Training CC staff with hygiene knowledge so that they can be empowered to contribute to the development of these standards would be important.

## Background

The postpartum period is an important time for women of Chinese ethnicity in Malaysia and elsewhere [1, 2]. Despite modernisation, most families still adhere strictly to a 30-day 'confinement period' known among the Chinese as 'zuo yue zi', with many do's and don'ts passed down from generations to generations [2, 3]. During the confinement period, women follow traditional practices to maintain the balance between the 'Yin' and the 'Yang'. It is believed that childbirth causes an imbalance between the 'Yin' and the 'Yang' and failure to restore this balance could potentially be detrimental to the mother's health [4]. According to the ancient Chinese philosophy, the 'Yin' and 'Yang' are attributes of all things or phenomena in the universe. They are of opposing characters. For example, 'Yin' refers to 'cold' while 'Yang' refers to 'warmth' [5]. Therefore, practices that assist the 'Yang' which is reduced during childbirth are emphasized. This includes ensuring that the mother 'keeps warm' by avoiding draughts and consuming a specially prepared confinement diet [4].

Many families engaged a traditional postpartum carer, known locally as a 'confinement lady' or 'yue sao' to stay in the new mother's home during the confinement period and assist the mother. The traditional postpartum carer is traditionally someone who is considered an expert in the necessary postpartum diet and practices. Their skills were probably obtained through experience rather than formal training [1, 6]. The traditional postpartum carer would move into the home for the entire duration of the confinement. She would usually be given her own room. Depending on the mother's preference and feeding choice, the baby might room-in with the traditional postpartum carer during the night or be with the mother. The traditional postpartum carer would have full access to the baby as her primary role would be to care for the baby in order for the new mother to 'rest'.

Over the last decade, confinement centres (CCs) where post-partum Chinese mothers could stay and observe traditional post-partum practices during their confinement period, have emerged as an alternative option. There are no published reports on why some mothers choose to go to CCs instead of opting for the traditional postpartum carer, but the reduced availability of the traditional confinement lady, and the increased availability of CCs could be a reason. CCs are private establishments, usually converted from residential or commercial properties, with rooms for mothers' accommodation. CC staff are generally women who are

familiar with the Chinese cultural confinement requirements and diet, similar to a traditional postpartum carer. Although some CCs do employ qualified nurses and midwives [6], others may employ untrained staff to help [7].

Concerns about hygiene in these centres arose when anecdotal reports suggested that babies in CCs were frequently hospitalised with serious infections. In 2007, Rai et al published a report about an Echovirus infection outbreak in a CC and poor hygiene practices in that CC were highlighted [8]. Therefore, to learn more about how hygiene is practiced in CCs, we asked mothers who had chosen to stay in a confinement centre about hygiene practices they had observed during the stay. Mothers who had employed a traditional postpartum carer to help them at home were also asked about the traditional postpartum carer's hygiene practice. This is part of a larger study where we looked at mothers' breastfeeding experiences in CCs and compared these with a cohort of women who had their traditional confinement period at home [7].

The aim of this paper is to describe the practice of hand hygiene observed in CCs, the availability of resources for hygiene and to determine the prevalence of health-related problems in CCs.

## Methods

Details of the study methods are published in the our primary paper on the breastfeeding experience of mothers staying in CCs [7] and we describe these in brief here. Malaysian mothers of Chinese ethnicity who intended to spend their confinement period in a CC, who had delivered term healthy infants and had the intention to breastfeed, were recruited prior to discharge from August to October 2017. For every mother that went to a CC, we also recruited a comparison mother, who was as far as possible the next woman from the same hospital who went home for her confinement period. Some of the mothers who went home engaged a traditional postpartum carer. For this paper, we used this comparison group to gauge hand hygiene practices in the CC with the traditional postpartum carer, and to compare health-related problems in CCs with those in the community. Recruitment and collection of demographic characteristics was done by the baby's attending doctor, who apart from this was not otherwise involved in the study. Written consent was obtained from the mothers prior to recruitment. Demographic characteristics which included mother's age, education level, gravida as well as her infant's sex, gestational age and birth weight, were collected using a self-administered questionnaire. To avoid any alteration in behaviour, mothers were not told at recruitment that they would be asked about hygiene practices.

After discharge, there was no contact between the research team and the mother until immediately after her 30-day confinement period. At this point we conducted a telephone interview with all mothers. Two recent graduates from our medical school were trained to conduct the interviews. In order to reduce variability, we designed a set of structured questions and the two interviewers followed these during the interview. The training involved an understanding of the data that is to be collected, familiarisation with the structured questions and mock interviews. During analysis we could not find any systematic differences in responses across interviewers.

We firstly asked mothers questions related to their baby's general health. We then categorised the reported health related problems to those that we judged were possibly related to infection and those that were probably not related to infections. Health problems possibly related to infection (HPRI) included fever, diarrhoea. Those that we judged to be health problems that were unlikely related to infection (HPUI) included neonatal jaundice and

regurgitation of feeds. If any health problem was reported, we asked if they had sought advice from a healthcare professional.

Where applicable, we asked whether or not they had observed their CC staff or traditional postpartum carer (in the case of those at home), washing their hands before handling their babies; and their response if hand-washing practices were not observed. We did not ask mothers staying at home who did not employ traditional postpartum carer whether or not their family members washed hands because the information was deemed to be possibly unreliable. Specifically for mothers who went to CCs, we asked the number of rooms in their CC, the number of mothers staying together in a room, if they had a hand basin in their room, if soap was available in all hand basins, what was provided for mothers to dry their hands, whether alcohol hand sanitizers were available, the number of babies in each nursery, availability of quarantine rooms for sick mothers or babies and their perception of cleanliness in their CC (either very dirty, somewhat dirty, clean or very clean). At the end of the interview, we asked mothers to share with us anything else they would like to about their experience in CCs. A sample of the interview questions (S1 File) can be found in the Supporting Information File.

All questions used had been tested in a separate group of mothers not involved in the study. The telephone interview was conducted by two trained research staff and the responses were directly entered into a specially designed interview form. A sample size of 188 was calculated based on the primary objectives of the primary study [7].

This study was registered with the National Medical Research Registry (NMRR-17-1174-36384 S1) and received ethical approval from the Joint Penang Independent Ethics Committee (JPEC 02-18-0026). All mothers gave written informed consent.

## Data analyses

We tabulated the demographics of the mothers according to place of confinement. Continuous data was presented as means with standard deviations (SD) and categorical data presented as frequency with percentage (%). Chi-square analysis was used to compare the baseline characteristics between mothers staying in confinement centres (CCs) and those staying at home. Responses from mothers when asked, "Is there anything else about your CC that you would like to share?" were tabulated and categorized into groups. Some of these free field responses were quoted as illustrations. We used simple logistic regression and multiple logistic regression after adjusting for clinically important confounders to determine the likelihood of HPRI and HPUI as a function of CCs. The results were presented as crude and adjusted odd ratios (cOR and aOR) with 95% confidence intervals (CI). Statistical analysis was done using Stata 13 [9]. We considered a p-value of less than 0.05 as significant.

## Results

A total of 187 mothers consented to participate, of which 88 (47%) stayed in a CC and 99 (53%) went home. At one-month post-partum, we were able to interview 77 (88%) mothers from the CC group and 96 (97%) from the home group. Based on the reported names of the CCs given by the mother, the 77 mothers in the CC group had gone to one of probably 27 CCs during their confinement period. Unfortunately, we were not able to verify reported names of CC because at the time of the study there was no record of all CCs in Penang available, and we are therefore uncertain about the exact number of CCs in the study. Of the 96 mothers from the home group, 34 hired a traditional postpartum carer while the remainder received care from family members (Fig 1).

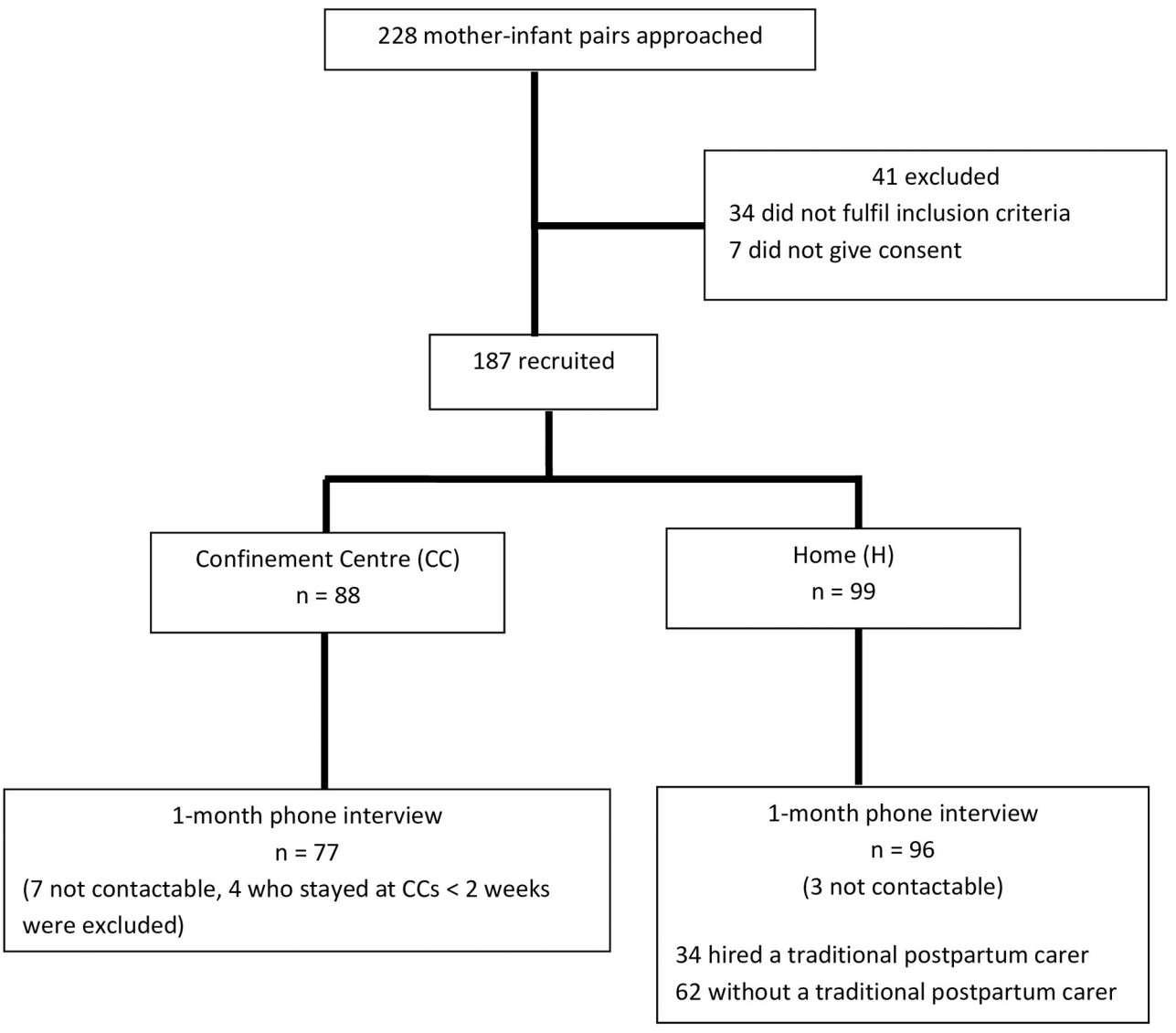

**Fig 1. Study flow diagram.** Flowchart of the cohort study showing the number of mothers who were recruited before discharge from the hospital after delivery of their baby, the number of mothers who went to a confinement centre or their own homes, and the number of mothers that completed the telephone interview a month after delivery. Some of the mothers who went back home hired a traditional postpartum carer to help during the confinement period.

### Demographic characteristics

The maternal and infant demographic characteristics are presented in Table 1. The overall mean maternal age was 32 (SD 4) years. Most mothers had tertiary education, and all had at least secondary school education, which reflects what is expected in Penang. The overall mean infant gestational age was 39 (SD 1) weeks and mean birth weight 3149 (SD 322) g. We found that significantly more primiparas went to CCs (53% CC vs 34% H, p = 0.01) but there were no differences in the age, education background, mode of delivery; infant gestation and birth weight between the two groups (Table 1).

**Table 1. Demographic characteristics of the mothers and infants (n = 187).**

| Characteristics | Place of confinement, n (%) | |
| --- | --- | --- |
| | Confinement centre (n = 88) | Home (n = 99) |
| Age of mothers (years), mean (SD) | 32 (4) | 32 (3) |
| Received tertiary education | 70 (80) | 80 (81) |
| Primigravida* | 47 (53) | 34 (34) |
| Male infant | 45 (51) | 56 (57) |
| Gestational age at birth (weeks), mean (SD) | 39 (1) | 39 (1) |
| Infant's birth weight (g), mean (SD) | 3141 (304) | 3156 (339) |

* p < 0.05

## Description of confinement centres

The description of the CCs came from mother's reports during the interview. More than one mother may have stayed in the same CCs. The CCs had between four to 10 rooms for mother's accommodation.

The number of mothers staying together in a room ranged from one to four. Forty-five mothers occupied a single bedded room, 20 mothers occupied a two-bedded room, 11 occupied three-bedded room and one stayed in a four-bedded room. Most of the mothers did not room-in with their babies (n = 71, 92%). Instead, their babies slept in the common nursery; and majority of mothers (n = 57, 74%) spent less than six hours a day with their babies. Regardless of CC size, all had a single common nursery for babies. The number of babies in the nursery at a time was reported to range from one to 17.

Of the mothers staying at home, 31 (32%) did not room-in with their babies, but only 11 (11%) spent less than six hours a day with their babies.

## Hygiene and infection control measures at confinement centres

When asked to rate the overall cleanliness of the CCs using a Likert scale of 0 to 3, with '0' being very dirty and '3' being very clean, all mothers reported that their centre was either 'clean' (n = 41, 53%) or 'very clean' (n = 36, 47%). However, only 17 (22%) mothers noticed that their CC staff washed hands in between handling babies and 33 (43%) mothers noticed that CC staff did not. When asked what they did if the CC staff failed to wash hands before handling a baby, two mothers reported that they went on to ask the staff to do so; two mothers said that they had not thought that this was something to be concerned about, and one just said that she felt sorry for the staff who was short-handed at that time. The remaining 27 (35%) mothers did not know if CC staff washed hands (Table 2).

Only 55 (63%) of mothers reported that their CCs supplied hand soap. Among the 32 (36%) mothers who reported that their CCs provided alcohol-based hand sanitizers, three reported that alcohol-based hand sanitizers were restricted to staff use only. Twenty-six (34%) mothers reported the availability of a sink for hand washing in their room. Only 23 (30%) mothers reported availability of hand towels for drying hands and some of these items were

**Table 2. Mothers' perception that hand hygiene was practised before handling babies.**

| | Hand hygiene practised | Hand hygiene not practised | Don't know if hand hygiene is practised |
| --- | --- | --- | --- |
| CC staff (n, %) | 17 (22) | 33 (43) | 27 (35) |
| Traditional postpartum carer (n, %) | 34 (35) | 6 (18) | 11 (32) |

**Table 3. Mothers' perception of the availability of hand hygiene resources at CCs (a total of 77 responses from 59 mothers at 26 CCs).**

| Hand hygiene resource | Available (n, (%)) |
|---|---|
| Hand basin in own room | 26 (34%) |
| Readily available hand soap at each sink | 55 (71%) |
| Hand towels to dry hands | 23 (30%) |
| Alcohol hand sanitizers | 32 (42%) |

reported to be either a single cloth-towel that was shared by everyone in the centre or toilet rolls (Table 3).

The availability of a quarantine room for sick mothers was reported by 17% of mothers while the availability of quarantine rooms for sick babies were reported by 24% of mothers. We do not have details on whether the quarantine rooms were meant for single or multiple users. One mother reported that her CC required all visitors to don gowns prior to entering the nursery. When the mothers were asked if there was anything else they would like to share with us, they revealed one or more of the comments related to poor hygiene listed in Table 4. These comments came from 31 mothers. It is likely that some of these mothers could have been in the same CC but we do not have information to determine what proportions of CCs had these issues.

With regards to hygiene practices by traditional postpartum carers at home, 6 (18%) of mother reported that their traditional postpartum carer did not wash hands before handling their baby and after changing diapers while 11 (32%) mothers did not know whether their traditional postpartum carer practiced hand hygiene. When we asked what they did when they saw poor hand hygiene, two reported that they asked their traditional postpartum carer to do so, while four did not do anything.

**Table 4. Comments related to hygiene in confinement centres.**

| Comments | Number of mothers who made this comment |
|---|---|
| Staff shortage and inexperienced staff who were unaware of hygiene practices | 9 |
| Only one toilet to be shared by all mothers hence quite dirty | 4 |
| A common towel used to burp all babies in the nursery | 2 |
| A common hand towel used by all mothers to wipe their hands | 2 |
| The same pail that was used for holding bath water was used for washing the floor | 1 |
| The same basin used to wash babies' bottoms was also used to wash milk bottles | 1 |
| Milk bottles that fell to the floor (staff fell asleep) were simply picked up and used to continue feeding the baby without being washed | 1 |
| Breast-pump parts were just soaked in hot water and not properly sterilized | 1 |
| Use of a common milk bottle that was sticky and dirty looking | 4 |
| Infrequent changing of diapers, cot sheets and bed sheets | 10 |
| Nursery cramped and not spaced | 4 |
| 3 babies sharing a single cot | 4 |
| Flies and mosquitos in their room | 2 |
| Poor quality paper hand towels—toilet rolls that easily disintegrate | 8 |
| Alcohol hand sanitizers were only for staff usage | 3 |

## Babies' general health at CCs and at home

Baby's general health at CCs and home were generally similar. HPRI were reported by 11 mothers (14%) from CCs compared to 13 mothers (14%) from home; p = 0.86. Of these, 10 mothers from CC compared to 13 mothers from home consulted a healthcare professional. Reported HPRI included one or more of these: 'fever', 'viral infection', 'cough', 'stuffy nose' 'runny nose' and 'oral thrush'. None of the infants from the CC group had any form of serious illness. One infant from the Home group was hospitalised for an unspecified viral infection which the mother thought was caught from the baby's older brother.

The main HPUI was 'jaundice'. Others included one or more of the following: 'colic', 'constipation', 'regurgitation' and 'umbilical hernia'. HPUI were reported by 70 mothers (92%) from CCs compared to 88 mothers (92%) from home; p = 0.92. Of these, 43 mothers from CC and 63 mothers from home consulted a healthcare professional.

Simple logistic regression showed no association between HPRI and place of confinement, OR 1.08 (95% CI 0.45 to 2.57), p = 0.86. There was also no association between HPUI and place of confinement, OR 1.06 (95% CI 0.35 to 3.20), p = 0.92. Multiple logistic regression adjusted for known clinically important confounders (maternal education level, spent less than six hours a day with baby, not sleeping with baby at night and no exclusive breastfeeding) also did not show that the CC or home was a factor affecting HPRI, aOR 1.28 (95% CI 0.36 to 4.49), p = 0.71; or HPUI, aOR 2.01(95% CI 0.52 to 7.82), p = 0.32 (Table 5).

When we asked mothers if they had anything else to share with us, we found three events that could indicate a possibility of infection transmission in CCs. One mother reported that there were visits from the health authorities to her CC because a number of babies (but not her baby) in her CC had fever and were hospitalized. One mother reported that all babies in her CC had either blocked or runny noses within a one-week period. Another thought the oral thrush and rashes on her baby were due to sharing of a baby wash cloth at the CC. The mothers who reported these events came from three different CCs.

## Discussion

The main finding from our study was inadequate hand hygiene and infection control facilities in CCs. Despite this, mothers reported that they were satisfied with their CC's cleanliness. Meanwhile, hand hygiene was also inadequate among the traditional postpartum carers for mothers staying at home. Due to small numbers, we were unable to show whether there was a difference for the type and prevalence of reported health problems between the two groups. Nevertheless, the report that a CC had visits from the health authorities could indicate that infection related events were occurring and this would be of concern. Unlike home postpartum care where the baby is cared for within his own family, CCs use a single nursery for many

**Table 5. Crude and adjusted ORs for HPRI and HPUI defined by place of confinement.**

| | | Number of mothers who reported (n) | Odds Ratio, OR (95% CI) | Adjusted Odds Ratio, aOR (95% CI) [a] |
|---|---|---|---|---|
| HPRI | Confinement Centres (n = 77) | 11 | 1.08 (0.45, 2.57) | 1.28 (0.36, 4.49) |
| | Home (n = 96) | 13 | | |
| HPUI | Confinement Centres (n = 77) | 70 | 1.06 (0.35, 3.20) | 2.01 (0.52, 7.82) |
| | Home (n = 96) | 88 | | |

HPRI: Health problems probably related to infections

HPUI: Health problems probably unrelated to infections

[a] Adjusted for maternal education level, spent less than six hours a day with baby, not sleeping with baby at night and no exclusive breastfeeding

babies and infections can easily spread from one to another. As stated in the introduction, we were aware of reported outbreaks and this was part of the motivation for this study.

Practices at CCs that may potentially have caused infection were noted. These practices most likely result from lack of resources, inconsistent hand hygiene practices and over-crowding. There was probably a lack of awareness both among the CC staff and mothers about the importance of hand hygiene. We were surprised to find 35% of mothers not cognisant of whether or not their CC staff washed hands before handling their babies.

We are unable to find other studies looking at confinement centres, but our study draws parallels with studies conducted with nursing homes and child-care centres. One similarity between CCs and nursing homes is that they have populations who are relatively susceptible to infection. These studies found that over-crowding and lack of hand hygiene led to infection transmission [10, 11]. A number of studies described how infection could be prevented through improving hand hygiene practices, the availability of resources and improved role modelling [12–16]. In addition, these studies also found that education and training could effectively increase hygiene practices in nursing homes [11, 16, 17]. Drawing from the findings of these studies, it is very likely that all of these could apply to CCs. Therefore, we could expect that if education and training were put in place, hygiene practices in CCs could improve. In addition, prior exploration of mothers' and CCs' staff understanding and knowledge about hygiene would be useful when designing training. However, good hand hygiene practices are known to be one of those practices that are difficult to sustain and therefore regular audit and feedback to improve sustainability might also be needed [18].

Current guidelines for hygiene practice in healthcare settings differ little in their recommendations but not much is known about the appropriate standard of care in community settings such as CCs. Infection control as it is practiced in healthcare settings may be difficult to implement in CCs and is costly. There is currently little research to guide practice. It is noted that CCs are not healthcare institutions, and their staff are not healthcare staff. In addition, the traditional confinement care offered by CCs is not a medical treatment but at the same time CCs need to be cognisant of the increased infection risk to neonates and have adequate infection prevention strategies in place.

Studies have shown that nursing homes struggle to strike a balance in attempting to preserve a homelike environment and hospital-level measures to control of infection [11, 19, 20]. This is likely to apply to the CC environment as well. CCs would need to consider what measures if implemented would be accepted by both staff and mothers and could be complied with. However, at the same time there is no doubt that infection control measures are needed and hand hygiene is obviously the place to start. Research in this area is much needed as well as research on effective training and methods of consolidating hand hygiene practice in CCs. In addition, since there were reports of shared equipment and feeding items, these should also be further pursued.

Many of the home-based traditional postpartum carers in our control group were also reported to not practice good hand hygiene. There is currently no literature about their hygiene knowledge and practices. However, to improve safe practices, home-based traditional postpartum carers should also be drawn into training interventions.

We found many mothers who stayed at CCs were discouraged from rooming-in with their babies, and their babies spent most of the time in the nursery [7]. This could partly be due to traditional belief of the need for mothers to rest but it could also be due to convenience of caring for all the babies in one place. Since there is a body of evidence showing that both mother-infant rooming-in and breastfeeding prevent infections [21–24], ways to improve these practices could also be looked at. Although exclusive breastfeeding rates in this study cohort were as good if not better that other local populations, most of the mothers were just feeding their

babies with expressed breastmilk and not breastfeeding directly from the breast [7]. Feeding expressed breastmilk carries an increased risk of infection since it involves use of breast pumps and bottles which need a high level of disinfection [25]. One way to improve direct breastfeeding would be to provide training to CC staff so that they can be empowered to provide support for direct breastfeeding and to provide rooming-in facilities for mother and baby. This might mean that maternal accommodation needs further studies to establish appropriate recommendations, for example, spacing between mothers. There is also a possibility of considering kangaroo care as a means of infection prevention, but studies are needed both in terms of feasibility and safety of practicing kangaroo care in a CC setting.

A limitation of our study would be that we did not have accurate data on which CC the mothers went to. We feel that our sample of mothers reasonably represents the mothers using CCs in Penang, however it is probable not all CCs in Penang were represented in the data. Since our sampling was of women and not CCs and these 77 women went to around 27 CCs, our data represents the number of women and their babies who were exposed to poor hygiene practices and not the number of CCs having poor hygiene practices. Another limitation of the study was the sample size which was not calculated to show a difference in HPRI rates. It is also important to note that our findings are those perceived by mothers.

The data was collected after they had left the CC and we did not verify their reports.

In addition, if for instance a hand basin was not perceived to be present, we were unable to determine which one of these 3 possibilities could be taking place. Once possibility was that hand basin was actually not present and the participant had actually looked for it and could not find it. Another possibility was that they were not even aware that a hand basin should be present and had not noticed it even if the hand basin was actually present (observer bias). The third possibility was that the hand basin was actually present, but they could not recall (recall bias).

To overcome these biases, further studies, perhaps using direct observation could be performed and should involve CC operators and managers. Therefore, there is a need to establish rapport with them early.

## Conclusion

We found unsatisfactory hygiene practices in CCs as reported by mothers who spent their confinement period there. We were not able to establish any direct evidence of infection transmission. However, based on a previous report [8] and the anecdotal reports given by mothers in this study along with the hygiene practices reported in this paper, it is likely to be happening. Therefore, future studies, including intervention studies, are urgently needed to establish an appropriate hygiene standard in community postpartum care facilities such as these, as well as the best method to implement this standard. Training CC staff to empower them with hygiene knowledge so that they can be involved and contribute to the development of these standards would be important.

## Supporting information

**S1 Data.**
(XLSX)

**S1 File.**
(PDF)

## Acknowledgments

The authors acknowledge Ru Jian Jonathan Teoh (initial draft of the protocol); Adele Tan, Hon Kit Cheang, Yee Chern Hwang, Giap Liang Dan, Jessica Tan, Kwai Meng Pong, Siti Khadijah Hamdan, Balkees Abdul Majeed (mother recruitment); Caryn Lim, Wei Wen Lee, Zcho Huey Lee, Claire Lee and Bee Hong Ang (data management); as well as all mothers and hospitals who participated in this study.

## Author Contributions

**Conceptualization:** Siew Cheng Foong, Wai Cheng Foong, May Loong Tan, Jacqueline Judith Ho.

**Data curation:** Siew Cheng Foong, Wai Cheng Foong, May Loong Tan.

**Formal analysis:** Siew Cheng Foong, Wai Cheng Foong, May Loong Tan, Jacqueline Judith Ho.

**Funding acquisition:** Siew Cheng Foong.

**Investigation:** Siew Cheng Foong.

**Methodology:** Siew Cheng Foong.

**Project administration:** Siew Cheng Foong.

**Resources:** Siew Cheng Foong.

**Supervision:** Siew Cheng Foong.

**Visualization:** Siew Cheng Foong.

**Writing – original draft:** Siew Cheng Foong, Wai Cheng Foong, May Loong Tan, Jacqueline Judith Ho.

**Writing – review & editing:** Siew Cheng Foong, Wai Cheng Foong, May Loong Tan, Jacqueline Judith Ho.

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
