## [Decision Letter · Decision Letter 0]

19 Jul 2021

PONE-D-21-03743

Mothers’ hygiene experiences in confinement centres: a cohort study

PLOS ONE

Dear Dr. Foong,

Thank you for submitting your manuscript to PLOS ONE. After careful consideration, we feel that it has merit but does not fully meet PLOS ONE’s publication criteria as it currently stands. Therefore, we invite you to submit a revised version of the manuscript that addresses the points raised during the review process.

Both reviewers have raised a number of concerns regarding the study design and data interpretation. Please ensure that, in responding to the reviews, you provide careful clarification of each of the points raised.

We look forward to receiving your revised manuscript.

Kind regards,

Jamie Males

Staff Editor

PLOS ONE

Journal Requirements:

2. Please including more information on the number of interviewers, their training and characteristics; and please provide the interview guide used.

Reviewers' comments:

Reviewer's Responses to Questions

**Comments to the Author**

1. Is the manuscript technically sound, and do the data support the conclusions?

Reviewer #1: Yes

Reviewer #2: Partly

2. Has the statistical analysis been performed appropriately and rigorously? 

Reviewer #1: No

Reviewer #2: Yes

3. Have the authors made all data underlying the findings in their manuscript fully available?

Reviewer #1: No

Reviewer #2: Yes

4. Is the manuscript presented in an intelligible fashion and written in standard English?

Reviewer #1: Yes

Reviewer #2: No

5. Review Comments to the Author

Reviewer #1: This paper explores hygiene conditions for women undergoing post partum confinement in homes and in confinement centres in Malaysia, with a focus on hand hygiene in the confinement centres. The researchers interviewed mothers over the telephone after the confinement.

Interesting paper and topic.

Major Comments:

Line 219: Can the authors include the exact questionnaire for the interviews as a supplementary file? Seeing the specific questions is very useful for interpreting the results.

Table 4 is perhaps the most interesting part of the manuscript and so I'd like to see a bit more detail provided on this part of the study. What was the question used to glean this information? Did the interviewer suggest any of these responses are did they all come from an open ended question with no prompting? Can you say something about the relative frequency of the different issues? I'd be particularly interested in knowing what proportion of the facilities had these issues.

My impression of some of the issues raised in table 4 is that they are both more likely to cause infection than failure to wash hands between babies and that many of them are much easier to fix than low hand hygiene compliance. E.g., contaminated bottles and towels, no bath tub to bathe the babies etc. After hearing these reports, do the authors remain confident that "However, at the same time there is no doubt that infection control measures are needed and hand hygiene is obviously the place to start" (line 308)? To me, these seem like low hanging fruit rather than the hand-washing. I'd appreciate reading the authors view on this in the discussion.

Regarding the differences between the confinement centres and home: I agree that the results presented may be due to a false positive, but I think that the authors also should be less confident that this is certainly the case (line 278). It may be the case that infection rates in the home are also quite bad. I think a sensitivity analysis might be warranted. What this would to is tell you what effect size you are capable of detecting with 80% power with the given sample size. Conducting a sensitivity analysis in G*power (free software) will tell you what odds ratio you can detect with 80% power in your study. If this suggests odds ratios far in excess of what you might reasonably expect based on other studies of hygiene and infant care, you know that your study was underpowered and that your conclusion of a false negative is at least a plausible interpretation.

Minor Comments:

A few additional lines explaining what the confinement period is and how widespread it is would be useful at line 60.

At line 68, might be worth noting the reasons women choose the confinement centres over the TPCs. Is there a cost difference, for example?

At line 92, provide more information about how the women were sampled. What was the sampling frame? Was the selection random?

At line 127, provide more information on the research staff who conducted the interviews. Interviewer effects are common in this kind of work (i.e., one interviewer tends to elicit different responses). Can the authors confirm that there were no systematic differences in the responses to the different interviewers?

Line 109: Can the authors explain why "baby being inactive" suggests infection: to a non-medically trained reader like myself, this is not obvious.

Table 3: include percentages so it matches the other tables.

I'm not sure it is reasonable to include the anecdotal evidence on lines 266: in statistical terms, observations like this have little ability to pick up on true differences are may be due to chance. Perhaps the editor has a view on this?

Thank you authors for this interesting read!

Reviewer #2: The manuscript included an interesting and important topic. The paper has potential to improve further if include some more results/information, if available and thereby discuss accordingly. The specific comments are as below:

Abstract:

1. Under methods section, need to mention clearly the type of interview conducted.

2. Line 38-39: Does it mean rest 88% reported satisfactory hand hygiene practice of their TPCs during caring their babies

3. Page 3, line 39: Under result section, need to specify what does refer to ‘unsatisfactory hand hygiene’

4. Page 3, line 51: Rather say ‘customized training’ in place of empowering CC staff

Background:

1. Page 4, line 61: For the international readers, please describe what is ‘Yin and Yang’

2. Page 4, line 63: Since TPC is not a standard term, consider to spell out throughout the paper instead of using abbreviation

Methods:

1. Page 5, line 97-98: What data collection tools was used to collect baseline data?

2. Page 5, line 107 and 109: Instead of using the word ‘judged’, consider to replace with ‘assessed’ since this is a judgmental word

3. Page 6, line 125: What type of interview guideline was used for the interview-structured questionnaire, semi-structured or in-depth interview guideline? Please mention clearly

Data analyses:

1. Page 7, line 138: ‘baseline characteristics’- need to describe under the method section what was the information/data collected for baseline

Results

1. Page 8, line 159: The sub-heading is ‘Baseline characteristics’. Is this baseline or demographic characteristics?

2. Page 9, line 175: Table title shall be demographic characteristics

3. Page 10, line 187: “Regardless of CC size, all had a single common nursery for babies”. Please include the range of CC accommodation. This will help to understand readers how many babies stayed in a nursery. In a CC some mothers could occupy single to three or multi-bedded rooms but it doesn’t give the scenario the size of a CC or # of babies stayed in the common nurseries in the CCs.

4. Page 10, line 190: ‘……only 11 spent less than six hours a day with their babies’. Consider present the characteristics of CCs and accommodation arrangements in a table format.

5. Page 10, line 194: ‘….their centre was either clean or very clean’….Was the question asked in Likert scale?

6. Page 10, line 195-196: Since mother were not aware about the hand hygiene behavior, there is a possibility of recall bias. Also need to say if any mother could not recall the HH practice properly.

7. Page 10, line 200-201: This is a big %. This also reflect somewhat the mothers’ perception to importance of the hand hygiene. While this recall bias is a limitation, it is also an important point of discussion

8. Page 10, line 202: In the title of the table 2, it mentioned ‘Mothers’ perception’…but the table doesn’t reflect any perception, rather it only reported CC staff’s HH practices observed/noticed by the mothers. Please correct the spelling in the 2nd and 4th column heading ‘practiced’

9. Page 11, line 210: What is the % of singles cloth towel for common use and the toilet rolls supplied by the CCs?

10. Page 11, line 212: The table doesn’t present any perception. Rather this is only the availability of hand hygiene resources reported by the mothers

In the 3rd row of the table 3, need to mention the hand sanitizer is only for staff use.

11. Page 11, line 215-216: Was the quarantine rooms for single mother or baby or for multiple but limited persons?

12. In the table 4, 3rd row, did it mean same bottle use to feed multiple babies?

13. Page 12, line 232: ‘….11 mothers did not know whether their TPC practices hand hygiene’... didn’t know or couldn’t remembers? Since this was the home setting, not clear what was the arrangement of mothers and babies staying, in the same room or different room?

Also check the typo ‘did not washed’ in line 231.

14. Page 13, line 242: ‘…. Hospitalized for viral infection…..’ Did the mother mention what was the infection?

15. Page 14, line 268: ‘…… because a number of babies in her CC had fever….’ Was her baby also included here?

Discussion

1. Page 15, line 280-282: In this sentence it discussed the lack of hand hygiene and other practices likely result from lack of resource, inconsistent hand hygiene practices…..’

I wonder you’ve explored mothers’ general perception of hygiene, cleanliness and any link between lack of hand hygiene practices/cleanliness and potential illness/infection.

Although some results presented mother’s perception of sharing items as cause of babies’ illness, the above-mentioned variables either not explored or presented. Therefore, would like to suggest to present those data, if available and then discussed in the discussion section.

2. Page 15, line 284-285: ‘One similarity with these is that they are populations…..’ Did you mean mothers at nursing homes and conferment centres? Consider to rephrase the sentence

3. Page 15, line 289-290: To design a compelling and effective educational training, understanding of mothers’ and caregivers’ perceptions and practices related to hygiene is crucial, which are still under explored in this study.

4. Page 16, line 300-303: Same as comment as earlier. To improve the risk perception and IC practices, caregivers’ perceptions and knowledge in CCs need to be assessed/identified to design an acceptable intervention.

5. Page 17, line 316-317: The authors pointed out that many mothers who stayed at CCs were discouraged from rooming-in with their babies, which also suggest for further research to understand the existing norms and ritual among this group at the neonatal period.

6. Page 17, line 321: In this sentence, not sure what does mean by the ‘direct breastfeeding rates’.

The following sentence indicate that the exclusive BF included the expressed breastmilk. It needs to be mentioned explicitly and also present in the results section.

7. Page 17, line 324-325: In this sentence, it discussed about empowering the CC staff. But earlier it mentioned that the CCs were discouraged from rooming-in mothers with babies… so at this point rather make the point to encourage/train CC’s for direct BF by mothers, so that they could also encourage mothers for direct BF.

8. Page 17, line 336: The authors mentioned that “It is also important to note that our findings are those perceived by mothers”… One major limitation of this data is mothers informed about the hand hygiene practice after one month from their memory. Moreover, during recruitment, mothers were not informed that they would be asked about their observation for HH practices of the caregivers to avoid any alteration. However, on the other hand it could be mothers’ recall bias to report about hand hygiene practices. Moreover, measuring hand hygiene events is always tricky which includes not only the overall hand hygiene practices but also critical times that would potential for pathogen transmission. The data also does not reflect the mothers’ general perceptions to hand hygiene and other infection control practices. Without knowing these it is difficult to conclude whether mothers feel the need for hand hygiene during feeding and handing neonates. Because there is a link between mother’s risk perceptions and reported hand hygiene behavior. If the data support the above-mentioned information, would recommend to include those. Otherwise need to mention as limitation.

9. Page 18, line 338: Include reference where discussed ‘observer bias’.

10. Page 18, line 339-340: Also direct observation may require

Conclusion

1. Page 19, line 348: Better to replace the word ‘Empowering’ with ‘Training’

2. Page 19, line 348-350: Revise the sentence

• In the figure/ diagram that described enrollment, need some correction. Insert the required spaces between words in the diagram. In some places 2nd bracket ‘{‘ was used, which is not an usual practice for a diagram/figure

• Overall English grammar and some spelling (specific style (American/British) allowed in this journal) need to be checked

6. PLOS authors have the option to publish the peer review history of their article (what does this mean?). If published, this will include your full peer review and any attached files.

Reviewer #1: No

Reviewer #2: No

---

## [Author Response · Author response to Decision Letter 0]

13 Sep 2021

Editor's comment

Authors' reply

Thank you. We have edited the manuscript to meet PLOS ONE’s style requirements. Please let us know if we have missed anything.

Editor's comment

Please including more information on the number of interviewers, their training and characteristics; and please provide the interview guide used.

Authors' reply

The interview guide has been added as a supplementary file. Information about the interviewers has been added in Line 125 to Line 130.

“Two recent graduates from our medical school were trained to conduct the interviews. In order to reduce variability, we designed a set of structured questions and the two interviewers followed these during the interview. The training involved an understanding of the data that is to be collected, familiarisation with the structured questions and mock interviews. During analysis we could not find any systematic differences in responses across interviewers.”

Editor's comment

We note that you have indicated that data from this study are available upon request. PLOS only allows data to be available upon request if there are legal or ethical restrictions on sharing data publicly. For information on unacceptable data access restrictions, please see http://journals.plos.org/plosone/s/data-availability#loc-unacceptable-data-access-restrictions.

Authors' reply

We have uploaded the anonymized data set necessary to replicate our study findings as a Supporting Information file. We have indicated this addition in Line 398 to Line 399.

Editor’s comment

Your ethics statement should only appear in the Methods section of your manuscript. If your ethics statement is written in any section besides the Methods, please delete it from any other section.

Authors' reply

We have removed it from the other section. Our ethics statement now only appears in the Method section in Line 157.

Editorial Office’s comment

 Please include a legend for figure 1.

Authors' reply

We have included a legend as follows:

“Flowchart of the cohort study showing the number of mothers who were recruited before discharge from the hospital after delivery of their baby, the number of mothers who went to a confinement centre or their own homes, and the number of mothers that completed the telephone interview a month after delivery. Some of the mothers who went back home hired a traditional postpartum carer to help her during the confinement period.”

Editorial Office’s comment

Please include captions for your Supporting Information files at the end of your manuscript, and update any in-text citations to match accordingly. Please see our Supporting Information guidelines for more information: http://journals.plos.org/plosone/s/supporting-information.

Authors' reply

We have included captions as follows: 

“A sample of the interview questions (S1. Interview) can be found in the Supporting Information File.” (Lines 150-151).

“A sample of the interview questions (S1. Interview) and the minimal anonymized data set (S2. Dataset) have been uploaded as Supporting Information files.” (Lines 398-399)

Reviewer #1

This paper explores hygiene conditions for women undergoing post partum confinement in homes and in confinement centres in Malaysia, with a focus on hand hygiene in the confinement centres. The researchers interviewed mothers over the telephone after the confinement. Interesting paper and topic.

Authors' reply

Thank you for your comments and suggestions to improve our manuscript. We have edited many sections of our manuscript. The line numbers indicating where the changes are made refer to the line numbers in the cleaned up revised manuscript (the copy without the "track changes")

Reviewer #1

Line 219: Can the authors include the exact questionnaire for the interviews as a supplementary file? Seeing the specific questions is very useful for interpreting the results.

Authors' reply

We have included this as a Supplementary file.

Reviewer #1

Table 4 is perhaps the most interesting part of the manuscript and so I'd like to see a bit more detail provided on this part of the study. What was the question used to glean this information? Did the interviewer suggest any of these responses are did they all come from an open ended question with no prompting? 

Authors' reply

We have made this line clearer by including text to inform readers that these answers were obtained without any prompting. 

“When the mothers were asked if there was anything else they would like to share with us, they also revealed one or more of the comments related to poor hygiene listed in Table 4.” (Line 239 to Line 241)

Reviewer #1

Can you say something about the relative frequency of the different issues? 

Authors' reply

Yes, we have added this into Table 4.

Reviewer #1

I'd be particularly interested in knowing what proportion of the facilities had these issues. 

Authors' reply

We do not know what proportion of the facilities had these issues. Several of these mothers could have gone to the same facility. We have added information in the text to clarify this. 

"These comments came from 31 mothers. It is likely that some of these mothers could have been in the same CC but we do not have information to determine what proportions of CCs had these issues." (Line 241 to Line 243)

Reviewer #1

My impression of some of the issues raised in table 4 is that they are both more likely to cause infection than failure to wash hands between babies and that many of them are much easier to fix than low hand hygiene compliance. E.g., contaminated bottles and towels, no bath tub to bathe the babies etc. After hearing these reports, do the authors remain confident that "However, at the same time there is no doubt that infection control measures are needed and hand hygiene is obviously the place to start" (line 308)? To me, these seem like low hanging fruit rather than the hand-washing. I'd appreciate reading the authors view on this in the discussion. 

Authors' reply

We think that these are likely isolated observations, and it is not low hanging fruit because we don’t know where this was happening and how frequently it was occurring. Our data suggests that hand hygiene is a common problem and therefore it is a good place to start. However, we agree that the reports of shared equipment and feeding need to be addressed and hence have added this sentence at the end of the paragraph.

“In addition, since there were reports of shared equipment and feeding items, these should also be further pursued.” (Line 337 to Line 338)

Reviewer #1

Regarding the differences between the confinement centres and home: I agree that the results presented may be due to a false positive, but I think that the authors also should be less confident that this is certainly the case (line 278). It may be the case that infection rates in the home are also quite bad. I think a sensitivity analysis might be warranted. What this would to is tell you what effect size you are capable of detecting with 80% power with the given sample size. Conducting a sensitivity analysis in G*power (free software) will tell you what odds ratio you can detect with 80% power in your study. If this suggests odds ratios far in excess of what you might reasonably expect based on other studies of hygiene and infant care, you know that your study was underpowered and that your conclusion of a false negative is at least a plausible interpretation. 

Authors' reply

We are not sure what false positive the reviewer is referring to because Lines 277 - 278 in the original document stated that there is no difference between CCs and home. 

For the current version, we have added this explanation to make it clearer. “Due to small numbers, we were unable to show whether there was a difference for the type and prevalence of reported health problems between the two groups.” (Line 294 to Line 296 of the current version).

In addition, the aim of the paper was to describe hygiene practices in CCs so we don’t think we would pursue this issue.

Reviewer #1

Minor Comments:

A few additional lines explaining what the confinement period is and how widespread it is would be useful at line 60.

Authors' reply

The background has been edited to include this information. 

"…to a 30-day ‘confinement period’ known among the Chinese as ‘zuo yue zi’, with many do’s and don’ts passed down from generations to generations. During the confinement period, women follow traditional practices to maintain the balance between the ‘Yin’ and the ‘Yang’. It is believed that childbirth causes an imbalance between the ‘Yin’ and the ‘Yang’ and failure to restore this balance could potentially be detrimental to the mother’s health. According to the ancient Chinese philosophy, the ‘Yin’ and ‘Yang’ are attributes of all things or phenomena in the universe. They are of opposing characters. For example, ‘Yin’ refers to ‘cold’ while ‘Yang’ refers to ‘warmth’. Therefore, practices that assist the ‘Yang’ which is reduced during childbirth are emphasized. This includes ensuring that the mother ‘keeps warm’ by avoiding draughts and consuming a specially prepared confinement diet." (Line 59 to Line 70)

Reviewer #1

At line 68, might be worth noting the reasons women choose the confinement centres over the TPCs. Is there a cost difference, for example? 

Authors' reply

We have also added this information in Line 83 to Line 86: “There are no published reports on why some mothers choose to go to CCs instead of opting for the traditional postpartum carer, but the reduced availability of the traditional confinement lady, and the increased availability of CCs could be a reason."

Reviewer #1

At line 92, provide more information about how the women were sampled. What was the sampling frame? Was the selection random? 

Authors' reply

As mentioned in Lines 91 - 98 of the original manuscript, details about how the women were sampled published in Foong 2020.

To address the reviewer’s concerns, we have added more information to our current text as follows:

“Malaysian mothers of Chinese ethnicity who intended to spend their confinement period in a CC, who had delivered term healthy infants and had the intention to breastfeed, were recruited consecutively prior to discharge from August to October 2017. For every mother that went to a CC, we also recruited a comparison mother, who was as far as possible the next woman from the same hospital who went home for her confinement period. Some of mothers who went home engaged a traditional postpartum carer” (Line 107 to Line 113 of the current manuscript)

Reviewer #1

At line 127, provide more information on the research staff who conducted the interviews. Interviewer effects are common in this kind of work (i.e., one interviewer tends to elicit different responses). Can the authors confirm that there were no systematic differences in the responses to the different interviewers? 

Authors' reply

We have added a line in the text to indicate this at Line 126 to Line 130.

"In order to reduce variability, we designed a set of structured questions and the two interviewers followed these during the interview. The training involved an understanding of the data that is to be collected, familiarisation with the structured questions and mock interviews. During analysis we could not find any systematic differences in responses across interviewers."

Reviewer #1

Line 109: Can the authors explain why "baby being inactive" suggests infection: to a non-medically trained reader like myself, this is not obvious. 

Authors' reply

Since there were actually no babies being reported as ‘inactive’ by mothers, we have decided to take this word out to avoid confusion to non-medically trained readers.

Reviewer #1

Table 3: include percentages so it matches the other tables. 

Authors' reply

Percentages are now included in Table 3.

Reviewer #1

I'm not sure it is reasonable to include the anecdotal evidence on lines 266: in statistical terms, observations like this have little ability to pick up on true differences are may be due to chance. Perhaps the editor has a view on this?

Thank you authors for this interesting read!

Authors' reply

We have reworded the sentence in the text as below:

“We were not able to establish any direct evidence of infection transmission. However, based on a previous report and the anecdotal reports given by mothers in this study along with the hygiene practices reported in this paper, it is likely to be happening.” (Line 383 to Line 386.)

Reviewer #2

The manuscript included an interesting and important topic. The paper has potential to improve further if include some more results/information, if available and thereby discuss accordingly.

Authors' reply

Thank you for your comments and suggestions to improve our manuscript. We have edited many sections of our manuscript. The line numbers indicating where the changes are made refer to the line numbers in the cleaned up revised manuscript (the copy without the "track changes")

Reviewer #2

The specific comments are as below:

Abstract:

1. Under methods section, need to mention clearly the type of interview conducted.

Authors' reply

We have added the information as below:

“After their 1-month confinement period, they were contacted for a structured telephone interview about their experience." (Line 22 to Line 23)

Reviewer #2

Line 38-39: Does it mean rest 88% reported satisfactory hand hygiene practice of their TPCs during caring their babies.

Authors' reply

No. Lines 38-39 of the original document mentioned that “18% of mothers who employed TPCs reported that their TPC had unsatisfactory hand hygiene practices.”

Of the remaining 82%, 32% didn’t know whether or not their TPC had satisfactory hand hygiene practices. To make this clear, we have edited the sentence to include the percentage of those that said that they did not know whether or not their TPC practised hand hygiene. 

"Of mothers who employed traditional postpartum carers, 32% did not know if their carer washed hands after changing diapers and 18% reported that their carer did not." (Line 38 to Line 40)

Reviewer #2

Page 3, line 39: Under result section, need to specify what does refer to ‘unsatisfactory hand hygiene’

Authors' reply

We have re-written this in Line 38 to Line 40: 

"Of mothers who employed traditional postpartum carers, 32% did not know if their carer washed hands after changing diapers and 18% reported that their carer did not."

Reviewer #2

Page 3, line 51: Rather say ‘customized training’ in place of empowering CC staff

Authors' reply

Empowering is more than customized training, although it might start with this. Empowering suggests that the CC staff would take ownership of the infection control.

We have edited the sentence as below:

“Training CC staff with hygiene knowledge so that they can be empowered to contribute to the development of these standards would be important.” (Line 51 to Line 52)

Reviewer #2

Background:

Page 4, line 61: For the international readers, please describe what is ‘Yin and Yang’

Authors' reply

We have now described this in Lines 61 to Line 70.

Reviewer #2

Page 4, line 63: Since TPC is not a standard term, consider to spell out throughout the paper instead of using abbreviation

Authors' reply

Thank you for the suggestion. We have now spelt this out in full.

Reviewer #2

Methods:

Page 5, line 97-98: What data collection tools was used to collect baseline data? 

Authors' reply

Baseline data was collected using a self-administered questionnaire. We have added this information in Line 118 to Line 121.

"Demographic characteristics which included mother’s age, education level, gravida as well as her infant’s sex, gestational age and birth weight, were collected using a self-administered questionnaire."

Reviewer #2

Page 5, line 107 and 109: Instead of using the word ‘judged’, consider to replace with ‘assessed’ since this is a judgmental word 

Authors' reply

We believe that the word “judged'' is more appropriate in this situation. Assessments would be based on criteria. Eg. we “assess” students in an exam.

Reviewer #2

Page 6, line 125: What type of interview guideline was used for the interview-structured questionnaire, semi-structured or in-depth interview guideline? Please mention clearly

Authors' reply

We have added this information in Line 126 to Line 130.

"In order to reduce variability, we designed a set of structured questions and the two interviewers followed these during the interview. The training involved an understanding of the data that is to be collected, familiarisation with the structured questions and mock interviews. During analysis we could not find any systematic differences in responses across interviewers."

Reviewer #2

Data analyses:

Page 7, line 138: ‘baseline characteristics’- need to describe under the method section what was the information/data collected for baseline

Authors' reply

We have added this information in Line 118 to Line 121.

“Demographic characteristics which included mother’s age, education level, gravida as well as her infant’s sex, gestational age, birth weight, were collected using a self-administered questionnaire.”

Reviewer #2

Results

Page 8, line 159: The sub-heading is ‘Baseline characteristics’. Is this baseline or demographic characteristics?

Authors' reply

We have edited the sub-heading to demographic characteristics in Line 185.

Reviewer #2

Page 9, line 175: Table title shall be demographic characteristics

Authors' reply

Thank you. Edited as suggested in Line 194.

Reviewer #2

Page 10, line 187: “Regardless of CC size, all had a single common nursery for babies”. Regardless of CC size, all had a single common nursery for babies”. Please include the range of CC accommodation. This will help to understand readers how many babies stayed in a nursery. In a CC some mothers could occupy single to three or multi-bedded rooms but it doesn’t give the scenario the size of a CC or # of babies stayed in the common nurseries in the CCs.

Authors' reply

The range of accommodation had been mentioned in Lines 186 -187 of the original document and we have kept this sentence in Line 199 to Line 200 of the current version 

“The CCs had between four to 10 rooms for mother’s accommodation.”.

In Line 206 to Line 207 of the current version, we have also mentioned that “The number of babies in the nursery at a time was reported to range from one to 17.” 

We do not have information on how many mothers there were at any one time, but we have information that there could be as many as 17 babies in a nursery. Since almost all were singletons, we assume there would the same corresponding number of mothers.

Reviewer #2

Since mother were not aware about the hand hygiene behavior, there is a possibility of recall bias. Also need to say if any mother could not recall the HH practice properly.s stayed in the common nurseries in the CCs

Authors' reply

We had already mentioned recall bias in Line 336 of the original document. To make this even clearer, we have made further elaboration in the current version. Meanwhile, ‘recall bias’ is part of it, but it is not just about recall bias. It also relates to whether the participants were aware of what to look for. 

" In addition, if for instance a hand basin was not perceived to be present, we were unable to determine which one of these 3 possibilities could be taking place. Once possibility was that hand basin was actually not present and the participant had actually looked for it and couldn’t find it. Another possibility was that they were not even aware that a hand basin should be present and had not noticed it even if the hand basin was actually present (observer bias). The third possibility was that the hand basin was actually present, but they could not recall (recall bias)." (Line 370-376)

Reviewer #2

Page 10, line 190: ‘……only 11 spent less than six hours a day with their babies’. Consider present the characteristics of CCs and accommodation arrangements in a table format 

Authors' reply

Since the accommodation arrangements are described by participants are actually only their perception, we feel that further detail could be misleading. Therefore, we have decided not to take up this suggestion.

Reviewer #2

Page 10, line 194: ‘….their centre was either clean or very clean’….Was the question asked in Likert scale?

Authors' reply

Yes, this was asked as a Likert scale.

We have added this information in Line 212 to Line 214.

“When asked to rate the overall cleanliness of the CCs using a Likert scale of 0 to 3, with ‘0’ being very dirty and ‘3’ being very clean, all mothers reported that their centre was either ‘clean’ (n = 41, 53%) or ‘very clean’ (n = 36, 47%).”

Reviewer #2

Page 10, line 195-196: Since mother were not aware about the hand hygiene behavior, there is a possibility of recall bias.

Authors' reply

We agree that there is a possibility of recall bias and had addressed this in the discussion of the original document in Lines 336 – 339. “The data was collected after they had left the CC and we did not verify their reports. Therefore, they could be subject to recall bias as well as observer bias.”

We have now also added more text into the discussion to further elaborate on this. "In addition, if for instance a hand basin was not perceived to be present, we were unable to determine which one of these 3 possibilities could be taking place. Once possibility was that hand basin was actually not present and the participant had actually looked for it and couldn’t find it. Another possibility was that they were not even aware that a hand basin should be present and had not noticed it even if the hand basin was actually present (observer bias). The third possibility was that the hand basin was actually present, but they could not recall (recall bias)." (Line 370 to Line 376)

Reviewer #2

Also need to say if any mother could not recall the HH practice properly.

Authors' reply

This had been presented in Lines 199 - 200 as well as in Table 2 of the original manuscript but we have further revised this sentence in the current manuscript.

“The remaining 27 (35%) mothers did not know if CC staff washed hands. (Table 2).“ (Line 219 to Line 220 of current manuscript)

Reviewer #2

Page 10, line 200-201: This is a big %. This also reflect somewhat the mothers’ perception to importance of the hand hygiene. While this recall bias is a limitation, it is also an important point of discussion.

Authors' reply

We agree and have added text in the discussion to address this. 

"There was probably a lack of awareness both among the CC staff and mothers about the importance of hand hygiene. We were surprised to find 35% of mothers not cognisant of whether or not their CC staff washed hands before handling their babies." (Line 304 to Line 306.)

Reviewer #2

Page 10, line 202: In the title of the table 2, it mentioned ‘Mothers’ perception’…but the table doesn’t reflect any perception, rather it only reported CC staff’s HH practices observed/noticed by the mothers.

Authors' reply

We left it as perception to make the point that different participants would have a different level of awareness of the presence of these items in their CC. We can’t actually tell whether the items listed were present from their retrospective report. It is not just about recall either. If a hand basin is not perceived to be present: 

1. was it because the it wasn’t actually present and the participant is quite sure of that (because they had looked for it)

OR

2. was it because they weren’t even aware that a hand basin should be present and if it was there they just didn’t notice it

OR

3. was it because they cannot recall? 

We have also included this into the text in Line 370 to Line 376.

Reviewer #2

Please correct the spelling in the 2nd and 4th column heading ‘practiced’.

Authors' reply

The word here is used as a verb and not a noun, hence we think it should remain as ‘practised’ and not ‘practiced’. We are using British English.

Reviewer #2

Page 11, line 210: What is the % of singles cloth towel for common use and the toilet rolls supplied by the CCs?

Authors' reply

This has been added to the Table 3 and Table 4.

We have considered hand towels to be either cloth or paper towels including toilet rolls.

Reviewer #2

Page 11, line 212: 

In the 3rd row of the table 3, need to mention the hand sanitizer is only for staff use.

Authors' reply

We have clarified this in the text in Line 224 to Line 226, and also in Table 4.

“Among the 32 (36%) mothers who reported that their CCs provided alcohol-based hand sanitizers, three reported that alcohol-based hand sanitizers were restricted to staff use only.”

Reviewer #2

Page 11, line 215-216: Was the quarantine rooms for single mother or baby or for multiple but limited persons?

Authors' reply

We do not have details about this. 

We have added a sentence stating, “We do not have details on whether the quarantine rooms were meant for single or multiple users.” In Line 237 to Line 238.

Reviewer #2

In the table 4, 3rd row, did it mean same bottle use to feed multiple babies?

Authors' reply

Table 4, 3rd row was a statement recorded verbatim by a participant. It meant that the CC used one common towel when burping different babies. The interviewer did not clarify with the participant if this meant a same bottle was used to feed multiple babies.

Reviewer #2

Page 12, line 232: ‘….11 mothers did not know whether their TPC practices hand hygiene’... didn’t know or couldn’t remembers? Since this was the home setting, not clear what was the arrangement of mothers and babies staying, in the same room or different room?

Authors' reply

Added to the background (Line 75 to Line 80)

“The traditional postpartum carer would move into the home for the entire duration of the confinement. She would usually be given her own room. Depending on the mother’s preference and feeding choice, the baby might room-in with the traditional postpartum carer during the night or be with the mother. The traditional postpartum carer would have full access to the baby as her primary role would be to care for the baby in order for the new mother to ‘rest’."

Reviewer #2

Also check the typo ‘did not washed’ in line 231.

Authors' reply

Thank you for spotting the typo. ‘washed’ changed to ‘wash’.

Reviewer #2

Page 13, line 242: ‘…. Hospitalized for viral infection…..’ Did the mother mention what was the infection?

Authors' reply

The mother just mentioned that the doctor said it was a viral infection.

We have added the word ‘unspecified’ into the sentence to make this clearer. (Line 262).

Reviewer #2

Page 14, line 268: ‘…… because a number of babies in her CC had fever….’ Was her baby also included here?

Authors' reply

No, her baby was not included.

Reviewer #2

Discussion

Page 15, line 280-282: In this sentence it discussed the lack of hand hygiene and other practices likely result from lack of resource, inconsistent hand hygiene practices…..’

I wonder you’ve explored mothers’ general perception of hygiene, cleanliness and any link between lack of hand hygiene practices/cleanliness and potential illness/infection.

Although some results presented mother’s perception of sharing items as cause of babies’ illness, the above-mentioned variables either not explored or presented. Therefore, would like to suggest to present those data, if available and then discussed in the discussion section.

Authors' reply

This is a great idea but unfortunately, we cannot draw any conclusions from our data. The participants that reported the illness might have been on the alert for the hygiene issues whereas participants in CCs where all was well might not have noticed the lack of hygiene.

Reviewer #2

Page 15, line 284-285: ‘One similarity with these is that they are populations…..’ Did you mean mothers at nursing homes and conferment centres? Consider to rephrase the sentence

Authors' reply

Rephrased sentence to “One similarity between CCs and nursing homes is that they have populations who are relatively susceptible to infection.” Line 308 to Line 310.

Reviewer #2

Page 15, line 289-290: To design a compelling and effective educational training, understanding of mothers’ and caregivers’ perceptions and practices related to hygiene is crucial, which are still under explored in this study.

 Page 16, line 300-303: Same as comment as earlier. To improve the risk perception and IC practices, caregivers’ perceptions and knowledge in CCs need to be assessed/identified to design an acceptable intervention.

Authors' reply

Thank you for this good suggestion. We have added this into the text in Line 317 to Line 318. “In addition, prior exploration of mothers’ and CCs’ staff understanding and knowledge about hygiene would be useful when designing training.”

Reviewer #2

Page 17, line 316-317: The authors pointed out that many mothers who stayed at CCs were discouraged from rooming-in with their babies, which also suggest for further research to understand the existing norms and ritual among this group at the neonatal period.

Authors' reply

For some strata in Chinese society, not rooming-in is the norm. We added a sentence to explain this. 

“This could partly be due to traditional belief of the need for mothers to rest but it could also be due to convenience of caring for all the babies in one place.” (Line 344 to Line 346).

Reviewer #2

Page 17, line 321: In this sentence, not sure what does mean by the ‘direct breastfeeding rates’.

The following sentence indicate that the exclusive BF included the expressed breastmilk. It needs to be mentioned explicitly and also present in the results section.

Authors' reply

Clarified the sentence with some additional text. 

“Although exclusive breastfeeding rates in this study cohort were as good, if not better that other local populations, most of the mothers were just feeding their babies with expressed breastmilk and not breastfeeding directly from the breast.” (Line 348 to Line 351)

Reviewer #2

Page 17, line 324-325: In this sentence, it discussed about empowering the CC staff. But earlier it mentioned that the CCs were discouraged from rooming-in mothers with babies… so at this point rather make the point to encourage/train CC’s for direct BF by mothers, so that they could also encourage mothers for direct BF.

Authors' reply

We have added into the sentence that providing training to CC staff will help empower them to provide support for direct breastfeeding.

“One way to improve direct breastfeeding would be to provide training to CC staff so that they can be empowered to provide support for direct breastfeeding and to provide rooming-in facilities for mother and baby.” (Line 352 to Line 355).

Reviewer #2

Page 17, line 336: The authors mentioned that “It is also important to note that our findings are those perceived by mothers”… One major limitation of this data is mothers informed about the hand hygiene practice after one month from their memory. Moreover, during recruitment, mothers were not informed that they would be asked about their observation for HH practices of the caregivers to avoid any alteration. However, on the other hand it could be mothers’ recall bias to report about hand hygiene practices. Moreover, measuring hand hygiene events is always tricky which includes not only the overall hand hygiene practices but also critical times that would potential for pathogen transmission. The data also does not reflect the mothers’ general perceptions to hand hygiene and other infection control practices. Without knowing these it is difficult to conclude whether mothers feel the need for hand hygiene during feeding and handing neonates. Because there is a link between mother’s risk perceptions and reported hand hygiene behavior. If the data support the above-mentioned information, would recommend to include those. Otherwise need to mention as limitation.

Authors' reply

Thank you. We have added text into the discussion to address this in Line 370 to 379.

“In addition, if for instance a hand basin was not perceived to be present, we were unable to determine which one of these 3 possibilities could be taking place. One possibility was that hand basin was actually not present and the participant had actually looked for it and couldn’t find it. Another possibility was that they were not even aware that a hand basin should be present and had not noticed it even if the hand basin was actually present (observer bias). The third possibility was that the hand basin was actually present, but they could not recall (recall bias).

To overcome these biases, further studies, perhaps using direct observation could be performed and should involve CC operators and managers. Therefore, there is a need to establish rapport with them early.”.

Reviewer #2

Page 18, line 338: Include reference where discussed ‘observer bias’.

Authors' reply

We are not sure why a reference is needed for observer bias but have added a short definition of it in brackets in the text in Line 373 to Line 376.

“Another possibility was that they were not even aware that a hand basin should be present and had not noticed it even if the hand basin was actually present (observer bias). The third possibility was that the hand basin was actually present but they could not recall (recall bias).”

We have also added some information about overcoming observer bias.

“To overcome this bias, further studies, perhaps using direct observation could be performed and should involve CC operators and managers.” (Line 377 to Line 379)

Reviewer #2

Page 18, line 339-340: Also direct observation may require.

Authors' reply

We are unsure what the reviewer means here. 

Reviewer #2

Conclusion

Page 19, line 348: Better to replace the word ‘Empowering’ with ‘Training’

Authors' reply

We have revised the sentence to incorporate this in Line 388 to Line 390.

“Training CC staff to empower them with hygiene knowledge so that they can be involved and contribute to the development of these standards would be important.”.

Reviewer #2

Page 19, line 348-350: Revise the sentence 

Authors' reply

We have revised the sentence as below: 

“Training CC staff to empower them with hygiene knowledge so that they can be involved and contribute to the development of these standards would be important.” (Line 388 to Line 390)

Reviewer #2

• In the figure/ diagram that described enrollment, need some correction. Insert the required spaces between words in the diagram. In some places 2nd bracket ‘{‘ was used, which is not an usual practice for a diagram/figure

• Overall English grammar and some spelling (specific style (American/British) allowed in this journal) need to be checked 

Authors' reply

We did not use any of such brackets and are unable to locate any such brackets being used, nor anything that needs spaces between the words in the figure. Grammar and spelling have been checked.

---

## [Decision Letter · Decision Letter 1]

6 May 2022

Mothers’ hygiene experiences in confinement centres: a cohort study

PONE-D-21-03743R1

Dear Dr. Foong,

We’re pleased to inform you that your manuscript has been judged scientifically suitable for publication and will be formally accepted for publication once it meets all outstanding technical requirements.

Kind regards,

Avanti Dey, PhD

Staff Editor

PLOS ONE

Additional Editor Comments (optional):

Reviewers' comments:

Reviewer's Responses to Questions

**Comments to the Author**

1. If the authors have adequately addressed your comments raised in a previous round of review and you feel that this manuscript is now acceptable for publication, you may indicate that here to bypass the “Comments to the Author” section, enter your conflict of interest statement in the “Confidential to Editor” section, and submit your "Accept" recommendation.

Reviewer #1: All comments have been addressed

Reviewer #3: All comments have been addressed

2. Is the manuscript technically sound, and do the data support the conclusions?

Reviewer #1: Yes

Reviewer #3: Yes

3. Has the statistical analysis been performed appropriately and rigorously? 

Reviewer #1: Yes

Reviewer #3: Yes

4. Have the authors made all data underlying the findings in their manuscript fully available?

Reviewer #1: No

Reviewer #3: Yes

5. Is the manuscript presented in an intelligible fashion and written in standard English?

Reviewer #1: (No Response)

Reviewer #3: Yes

6. Review Comments to the Author

Reviewer #1: All my points have been addressed. Thank you authors for this interesting work. Look forward to seeing more research on this area in future.

Reviewer #3: This article is substantially improved from its original version. Cultural considerations regarding Confinement Centers is a fascinating topic which is under researched. I think it is good to have this somewhat straightforward and simple topic about hand hygiene published. But there is clearly so much for future research, including not only links to breastfeeding (which the authors have already published on and which they reference), but also to larger questions about cultural considerations regarding postpartum care in general. The current paper also references their other paper on breastfeeding for the methods, and I think the authors could talk more about their methods in this article as well. I appreciate they did not want to repeat what is already published, but the reader would benefit from additional methods details which are published in their breastfeeding paper. Although I do think the reader does benefit from reading the author's other paper as well. This article is a well written and on a fascinating topic, albeit framed in a very specific way, and I believe should lead to future research on cultural considerations regarding postpartum care.

7. PLOS authors have the option to publish the peer review history of their article (what does this mean?). If published, this will include your full peer review and any attached files.

Reviewer #1: No

Reviewer #3: **Yes: **Dr Tanya M Cassidy

---

## [Editor Report · Acceptance letter]

13 May 2022

PONE-D-21-03743R1 

Mothers’ hygiene experiences in confinement centres: a cohort study 

Dear Dr. Foong:

I'm pleased to inform you that your manuscript has been deemed suitable for publication in PLOS ONE. Congratulations! Your manuscript is now with our production department. 

Kind regards, 

on behalf of

Dr. Avanti Dey 

Staff Editor

PLOS ONE